# Reproducibility, and repeatability of corneal topography measured by Revo NX, Galilei G6 and Casia 2 in normal eyes

**Adam Wylęgała** [1,2] *, **Robert Mazur** [1,2], **Bartłomiej Bolek** [1,2], **Edward Wylęgała** [1,2]

**1** Ophthalmology Departament, Railway Hospital, Katowice, Poland, **2** School of Medicine, Division of Dentistry, Zabrze Medical University of Silesia, Katowice, Poland

* Adam.wylegala@gmail.com

**Data Availability Statement:** https://data.mendeley.com/datasets/kvs6258sdp/draft?a=0f60c172-fbcc-4185-904f-b6866a939314

## Abstract

### Purpose

To test the repeatability and reproducibility of the topography module in posterior segment spectral domain optical coherence tomography with Revo NX (new device) and to compare keratometry values obtained by a Scheimpflug tomography (Galilei G6) and a swept source OCT (Casia 2).

### Methods

In this prospective study, healthy subjects with nonoperated eyes had their central corneal thickness (CCT), anterior and posterior K1/K2 corneal power measured with the new device. Two operators made 6 measurements on the new device to check intraobserver repeatability and reproducibility, and measurement on Casia 2 and Galilei G6. Bland-Altman plots were used to assess the agreement between the devices for each analyzed variable.

### Results

94 eyes (94 patients) were studied. All devices produced significantly different mean CCT, the highest for Galilei 569.13±37.58 μm followed by Casia 545.00 ±36.15 μm and Revo 537.39±35.92 μm. The mean anterior K1 was 43.21 ± 1.37 for Casia 2 43.21 ± 1.55 for Revo NX and 43.19 ± 1.39 for Galilei G6, and the differences were insignificant p = 0.617. The posterior K1 for Revo NX was -5.77 ± 0.25 whereas for Casia 2 it was -5.98±0.22 and for Galilei G6–6.09±0.28 D p< 0.0001. The Revo NX showed intraclass correlation coefficient ranging from 0.975 for the posterior K2 surface, and 0.994 for anterior K1 and 0.998 for CCT.

### Conclusions

Revo NX is independent of the user and offers a high level of repeatability for the anterior and posterior cornea. The wide range of differences between the devices suggests they should not be used interchangeably.

**Funding:** Optopol Technology Ltd. provided the Revo NX equipment with corneal topography module used in this study. AW received a speaker's honorarium form Carl Zeiss and works as a consultant for Carl Zeiss Meditec. The funders had no role in study design, data collection and analysis, decision to publish, or preparation of the manuscript.

**Competing interests:** Optopol Technology Ltd. provided the Revo NX equipment with corneal topography module used in this study. AW received a speaker's honorarium form Carl Zeiss and works as a consultant for Carl Zeiss Meditec. Aw has a patent pending for the OCT angiography algorithm. Polish patent office P.418979. This does not alter our adherence to PLOS ONE policies on sharing data and materials.

## Introduction

The most important refractive part of the optic system of an eye is the cornea. Due to the difference between two refractive indexes of the air and tear film it has the highest refractive power of all the structures in the eye. Measuring the corneal topography is hence of major importance before performing cataract and refractive procedure, or monitoring the progression of a disease [1–3]. Furthermore, measurements of central corneal thickness are vital in diagnosis of glaucoma, Fuchs endothelial cell dystrophy and corneal graft rejection. Modern tomographers can measure many parameters including: anterior and posterior curvatures, pachymetry, refractive power, corneal thickness and provide high quality images. The introduction of Optical Coherence Tomography (OCT) to ophthalmology allowed a new way to quantify and visualize structures in anterior and posterior segment of an eye.

Anterior segment OCT utilized a coherence interferometer to generate 2 or 3 dimensional images of the anterior segment of the eye [4]. Currently there are two different types of OCT devices which allow to observe anterior segment. Devices such as Casia-2 are dedicated to imaging and analyzing the anterior segment only. While, devices like Zeiss Cirrus OCT, designed to obtain images of the posterior segment, are also capable of imaging the anterior segment after an optional anterior module is attached [5]. Revo NX is the latter type. Contrary to most OCT machines with add on lens that measure only anterior curvature, [6] it is cable of generating anterior and posterior corneal surface maps with respective keratometry. Measuring posterior corneal power is vital in keratoconic patients and in IOL calculation [7]. The major benefit of using a combined system is the lower price, and higher resolution. While drawback is the lack of collimated light at the cornea, whcihc leads to the messurements being distance dependent. Further the field of view is twice smaller then in the single use device.

There are two forms of measuring the precision of a device: repeatability and reproducibility. Repeatability means variability of results measured in short intervals, while reproducibility is defined as variability of results measured under different circumstances e.g., exams taken by different operators [8]. Accurate quantification of corneal power is of utmost importance in the age of premium IOL, and refractive surgery. Although pachymetry assessment can be a way to monitor corneal edema or to adjust IOP for corneal thickness, some previous studies that compared older devices concluded that the corneal parameters produced by other devices should not be used interchangeably. In this study, we evaluated the correlation and efficiency of measurements of the anterior segment of healthy eyes taken with the three devices.

The goal of this paper is to assess both the repeatability of the spectral domain OCT—Revo NX, and the agreement between a rotating Scheimpflug camera (Galilei Z6) and Anterior segment Swept Source OCT—Casia 2.

## Methods

This study was approved by the bioethical committee of the Silesian Medical University and adhered to the tenets of the Declaration of Helsinki. Before the examination, the participants had signed informed consent and had been informed of the experimental procedure. We included 94 eyes of 35 males and 59 females aged 32.34 ±10.21 in this prospective study. Subjects were students, interns, and workers of the hospital with no corneal conditions including ectatic disease such as KC. Recruitment time started in September 2018 and lasted till the end of January 2019. Participants who had been wearing any type of contact lenses less than 72 hours prior to the measurements were not included in the study, nor were those who had undergone any ophthalmic surgery e.g. cataract or refractive.

## Study devices

Revo NX, software version 9.0.0 (Optopol Technology Ltd, Zawiercie, Poland) is a high speed 110 000 A-scan/sec spectral-domain OCT operates at 830 nm center wavelength, with 5 μm axial and 18 μm transverse imaging resolutions. It can visualize the posterior segment of the eye and measure the axial length with an add-on lens as well as create maps of the cornea and images of the anterior segment. The device automatically acquires 16 B-scans of the 8 mm corneal diameter. Keratometry values in this study were calculated in the 3 mm central zone. Anterior, Posterior and True Corneal power, CCT is averaged from the central 3 mm zone. The device uses a refractive index of 1.3375 in order to convert the radius calculation expressed in mm to curvature power in D. To calculate the posterior K 1.336 and 1.376 refraction indexes are used.

The Galilei G6 Dual Scheimpflug Analyzer (Ziemer, Port, Switzerland) combines 20 Placido rings based topography with a dual rotating Scheimpflug camera 9. Scheimpflug technology is considered gold standard in corneal meassurements. Simulated keratometry (SimK) is calculated from the 0.5 to 2.0 mm annular (semichord) zone and is represented as dioptres using a refractive index of 1.3375. The posterior Mean K is calculated using a refractive index of 1.376 for the cornea and a refractive index of 1.336 for the aqueous humor. It is calculated over an area of 4 mm in diameter (2 mm radius or semichord).

A different technology is used by CASIA2 (Tomey Corporation, Nagoya, Japan) Swept Source anterior segment OCT (AS-OCT). It uses a swept laser 1310 nm wavelength, which is longer than in SD-OCT devices providing higher penetration but lower resolution, 50 000 A-scan/sec high-speed detector, and contrary to SD-OCT it lacks spectrometer. It uses a CMOS camera instead. Corneal power is calculated using a 1.3375 refractive index. Further, keratometry values are calculated on a 3.2 mm diameter.

## Measurement technique

All devices were placed in one darkened room. All measurements were taken on the same days by two trained operators. One eye of each subject was randomly chosen. Every participant had 6 Topo scan measurements on Revo NX (3 scans carried out by each operator), to measure repeatability and reproducibility, followed by one corneal map measurement on Galilei G6, Corneal Map scan on Casia 2. For every device, anterior and posterior K1 and K2 values were recorded as well as apical CCT. Only measurements well centered and with high-quality indexes were included in the study.

## Statistical analysis

Statistical analysis was conducted—using Statistica software ver. 13.1 (Dell Inc, Tulsa, OK, USA.) releases by Statsoft (Krakow, Poland). Numerical results for repeatability and reproducibility contain six quantities computed for observers separately and respectively for the entire dataset: mean, standard deviation (SD.), within-subject standard deviation (Sw.), test-retest repeatability (TRT.), within-subject coefficient of variation (CoV.), intraclass correlation coefficient (ICC.) were calculated for repeatability and reproducibility of the Revo NX. A comparison between 3 devices was analyzed using Bland-Altman plots. The normality of the data was tested using the Shapiro-Wilk test. The paired Student t-test was used to assess the differences between the devices. Statistical data in the form of an excel spreadsheet as well as a detailed description with the mathematical equation used will be available in Mendeley data depository from 24-MAY-2019 https://data.mendeley.com/datasets/kvs6258sdp/draft?a=0f60c172-fbcc-4185-904f-b6866a939314

## Results

### Interoperator repeatability and reproducibility

The operator impact on the device was insignificant with an interoperator intraclass correlation coefficient for both anterior and posterior K1 and K2 parameters ranging from 0.975 to 0.994 (Fig 1 and Table 1).

The Revo NX showed a high level of reproducibility that was also statistically insignificant with intraclass correlation coefficient ranging from 0.977 to 0.991 and standard deviation from 0.23 D for posterior K1 to 1.55 D for anterior K2. The intraoperator difference in the standard deviation in Anterior K1 was 0.01 and 0.02 for K2 while the posterior K1 and K2 standard deviation showed no difference (Fig 1 and Table 2). CCT showed even higher level of ICC of 0.998, with the mean CCT of 530.89±32.55 μm.

### Comparison

Differences in mean anterior K1 corneal measurements between G6 43.19 ± 1.39. Casia2 43.21 ± 1.37 and Revo NX 43.21 ± 1.55 were statistically insignificant (Fig 2). However the analysis showed statistically significant differences between anterior K2 for Casia 2. 44.17 ± 1.38, Revo NX 43.98 ± 1.53 and Galilei G6 44.15 ± 1.37 which were significant between Casia 2 and Revo p< 0.000, and Revo and G6 p = 0.004 (Fig 2). Differences between anterior keratometry of Casia 2 and Galilei G6 showed no significance with p = 0.21 and p = 0.46 for the K1 and K2 respectively. The devices were not interchangeable for the measurement of posterior K1 and K2. Posterior K1 showed significant differences between Galilei G6–6.09 ± 0.28. Revo -5.77±0.25 and Casia -5.98±0.22 for all comparisons p< 0.0001 (Table 3). The mean posterior K2 was -6.03 ±0.27 for Revo NX and -6.29 ±0.24 for Casia 2 and -6.53±0.39 for Galilei p< 0.0001 (Fig 3). The highest mean apical CCT was noted by Galilei G6 it was 569.13±37.58 followed by Casia 545.00 ±36.15 while Revo NX demonstrated the smallest CCT of 537.39 ±35.92 (Table 4). All comparisons were significant p<0.0001 (Fig 4).

## Discussion

In clinical medicine. the measurements performed in vivo are constantly changing and their true value is unknown. If a new method or a new device is brought to the market it is compared with the current well-established methods–the so called gold-standard. The changes between the current and a new method cannot be too big to influence the clinical decision. Bland and Altman proposed a graphical plot that is easy to interpret to determine the usefulness of a new method [8,9].

In this study. we compared the repeatability and interoperator reproducibility of a new corneal topographer module of Revo NX SD-OCT with Galilei G6 Schimpflug camera and Casia 2 SS-OCT in normal eyes. As it was concluded in many previous studies the measurements from two keratometric systems cannot be used interchangeably [3,10–12] There are two types of devices capable of measuring anterior and posterior keratometry: OCT based systems and Scheimpflug camera. Some OCT systems use swept-source technology featuring lower resolution but faster acquisition rate [13]. Others relay on spectral domain producing a smaller acquisition window but with a higher image quality [14]. The biggest advantage of AS-OCT over a Scheimpflug based system is that the numeric values are accompanied by the presence of high quality images that are superior in terms of resolution [15].

Crawford et. al compared Galilei. another Scheimpflug camera Pentacam (Oculus. Weltzar. Gemany) and Orbscan II (Bausch&Lomb. Rochester. USA). The authors showed a good level of repeatability. while Galilei exhibited best reapeatability [12]. Similarly. Meyer and his

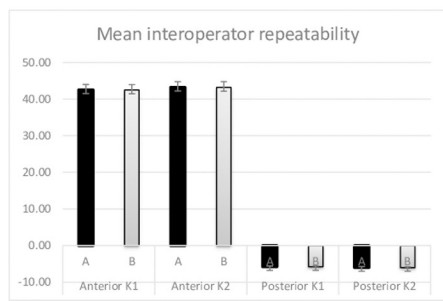 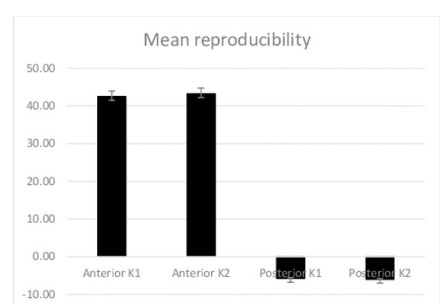 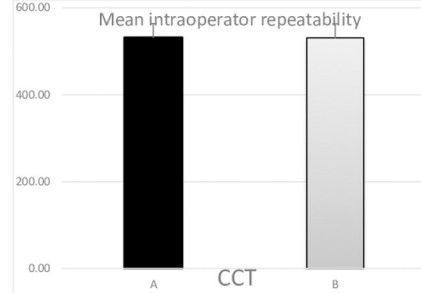

**Fig 1. Mean interoperator repeatability between operator A and B and reproducibility of Revo NX.**

colleagues compared Orbscan II. Galilei and Pentacam in keratoconic eyes and observed that Orbscan II has the least repeatable measurements. Furthermore there was no significant difference between Pentacam and Galilei [11]. Another study that showed no agreement between corneal diameters measured by Galilei. Orbscan and EyeSys (EyeSys Corneal Analysis System. Houston. Tx. USA) was published by Salouti et al. The authors concluded that these differences come from different measurement methods [16]. This is further complicated because the manufactures rarely disclose the method of capturing measurements. Kannengießer evaluated IOL topographies using Casia. Pentacam and TMS-2N (Tomey. Nagoya. Japan) and concluded that Casia creates a high level of variation compared with the other machines [17].

Casia showed higher dioptric values compared to Pentacam in both anterior K1 and posterior surface [18]. The authors speculate that these changes are due to the presence of various methods applied in the devices. As we showed in Table 3. Casia 2 demonstrated higher keratometry values compared to Scheimpflug device while Revo NX showed higher values only in anterior K2. Repeatability values in a similar study were 0.61%. 0.82%. and 0.80% for the SD-OCT. Pentacam. and ultrasound respectively [19]. Furthermore Savini et al. showed high agreement between videokeratographs and Scheimpflug device. However the level of agreement was considerably high around 1 D [20].

CCT was shown to be the highest in Scheimpflug device which is consistent with the previous studies. One study reported mean difference of 13.6 μm between Pentacam and Casia [18]. Another study found that for Pentacam and Casia the mean CCT was 544 μm and 533 μm

**Table 1. Intraoperator repeatability of Revo NX, each operator performed three measusrements.**

| Parameter | Operator | Mean | SD. | SW | TRT. | CoV.[%] | ICC. |
|---|---|---|---|---|---|---|---|
| Anterior K1 | A | 42.50 | 1.49 | 0.12 | 0.33 | 0.28 | 0.994 |
| | B | 42.47 | 1.50 | 0.14 | 0.39 | 0.33 | 0.991 |
| Anterior K2 | A | 43.19 | 1.54 | 0.16 | 0.46 | 0.38 | 0.989 |
| | B | 43.17 | 1.56 | 0.14 | 0.40 | 0.33 | 0.992 |
| Posterior K1 | A | -5.75 | 0.23 | 0.03 | 0.10 | -0.60 | 0.978 |
| | B | -5.76 | 0.23 | 0.03 | 0.09 | -0.56 | 0.981 |
| Posterior K2 | A | -6.00 | 0.26 | 0.04 | 0.11 | -0.63 | 0.979 |
| | B | -6.01 | 0.26 | 0.04 | 0.11 | -0.69 | 0.975 |
| Central corneal thickness | A | 531.05 | 32.54 | 1.47 | 4.07 | 0.28 | 0.998 |
| | B | 530.73 | 32.68 | 1.50 | 4.16 | 0.28 | 0.998 |

SD. = Standard deviation SW. = within-subject standard TRT. = test-retest repeatability, CoV. = within-subject coefficient of variation, ICC. = intraclass correlation coefficient,

**Table 2. Revo NX reproducibility based on six measurement from both operators.**

| Parameter | Mean | SD. | SW. | TRT. | CoV.[%] | ICC. |
|---|---|---|---|---|---|---|
| Anterior K1 | 42.48 | 1.49 | 0.14 | 0.40 | 0.34 | 0.991 |
| Anterior K2 | 43.18 | 1.55 | 0.16 | 0.46 | 0.38 | 0.989 |
| Posterior K1 | -5.75 | 0.23 | 0.04 | 0.10 | -0.61 | 0.977 |
| Posterior K2 | -6.00 | 0.26 | 0.04 | 0.11 | -0.65 | 0.978 |
| Central corneal thickness | 530.89 | 32.55 | 1.62 | 4.47 | 0.30 | 0.998 |

SD. = Standard deviation SW. = within-subject standard TRT. = test-retest repeatability, CoV. = within-subject coefficient of variation, ICC. = intraclass correlation coefficient,

respectively [10]. In our study the mean CCT measured by Revo NX was 537.39 ±35.9. 545.56 ± 35.7μm and 569.37± 37.0 μm for Casia and Galilei G6 respectively. Another work examined the comparison and repeatability between AS-OCT and Scheimpflug based system. It was discovered that the mean CCT was highest in ultrasound device. followed by Scheimpflug based and SD-OCT. Interoperator reproducibility was lowest in ultrasound based technology. The authors link the highest ultrasound thickness with the tear film dislocation partially caused by the anesthetic drops. Sheimpflug image system compared with the SS-OCT tends to

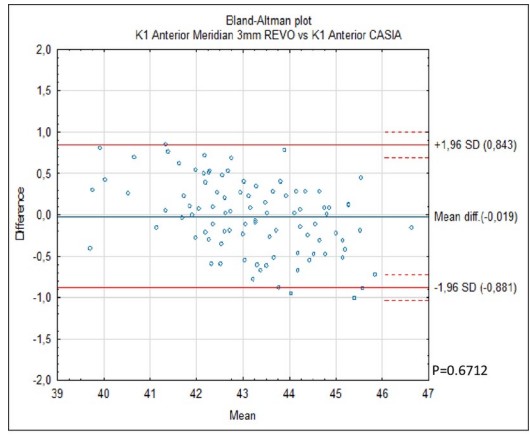
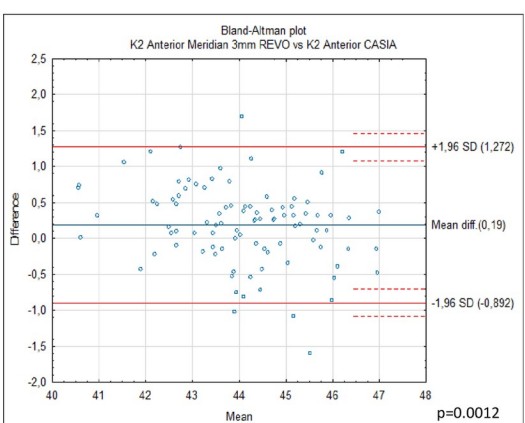
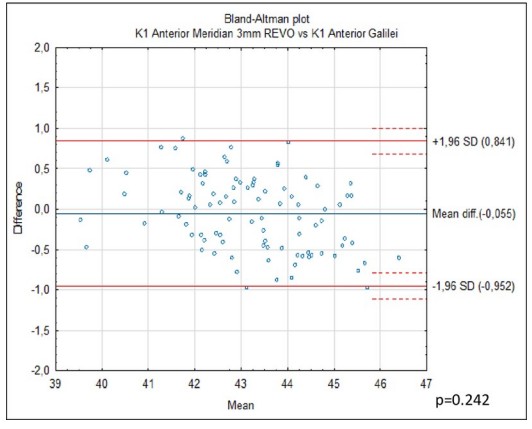
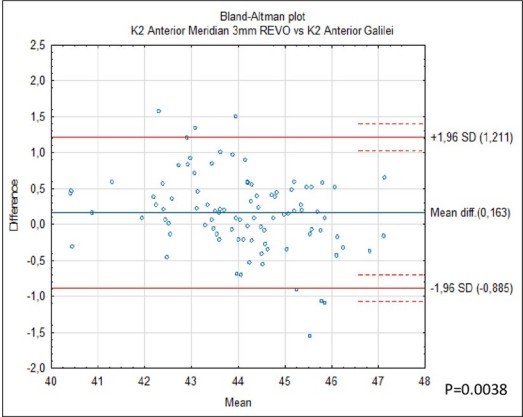

**Fig 2. Bland-Altman plots showing the agreement between anterior K1 obtained by the Galileli G6.** Casia 2 and Revo NX and K2 in 94 normal eyes. The mean difference is represented by the solid blue line whereas the dotted lines represent±1.96 SD.

**Table 3. Differences between mean of the Galilei G6 and standard deviations for the difference of CCT.**

| Value / Device | Galilei G6 | | Casia 2 | | Revo Nx | |
|---|---|---|---|---|---|---|
| | Mean | SD. | difference between G6 | SD. for the difference | Difference vs G6 | SD. for the difference |
| CCT | 569.13 | 37.02 | 23.93 | 9.58 | 31.74 | 10.32 |
| K1 anterior | 43.15 | 1.39 | -0.04 | 0.28 | -0.06 | 0.46 |
| K2 anterior | 44.15 | 1.38 | -0.03 | 0.35 | 0.16 | 0.53 |
| K1 posterior | -6.10 | 0.28 | -0.11 | 0.21 | -0.33 | 0.24 |
| K2 posterior | -6.53 | 0.39 | -0.23 | 0.29 | -0.50 | 0.31 |

Difference was always calculated (Galilei G6)–(Casia 2 or Revo Nx). SD.-standard deviation.

provide higher CCT values [19]. The reason why Scheimpflug produced the highest CCT is due to the probable inclusion of tear film into CCT [10]. Different methods yield different results due to the variuose reference models used such as average speed of sound or refractive index.

Previous studies compared the agreement between older types of devices. in our study we related the latest version of swept source OCT and dual Scheimpflug combined with placido

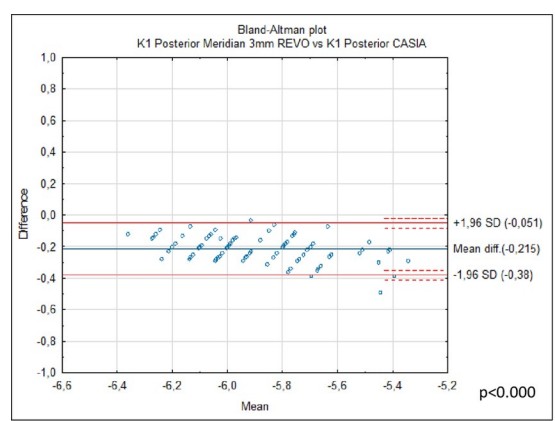

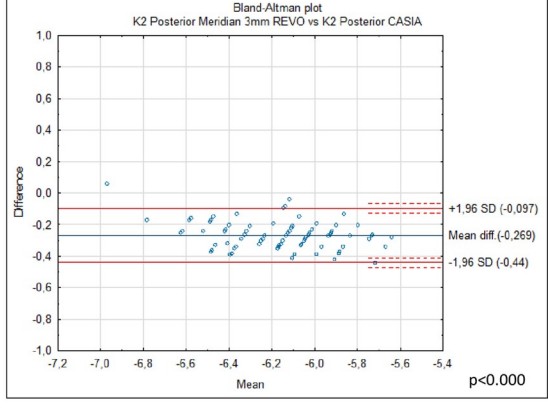

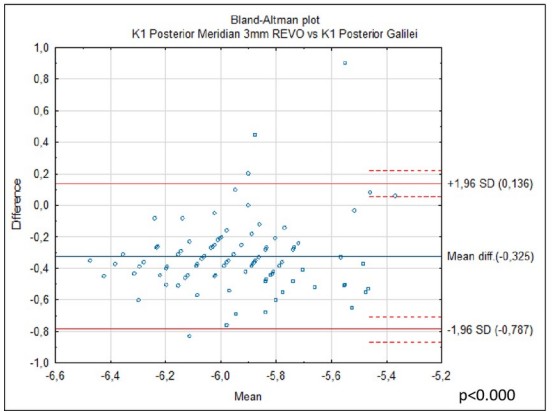

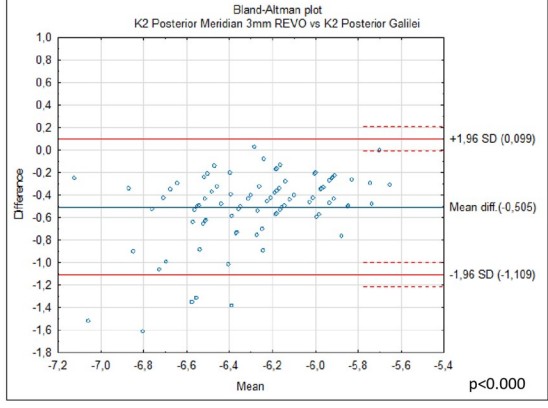

**Fig 3. Bland-Altman plots showing the agreement between posterior K1 and K2 obtained by the Galileli G6 Casia 2 and Revo NX in 94 normal eyes.** The mean difference is represented by the solid blue line whereas the dotted lines represent ±1.96 SD.

**Table 4. The mean, difference, range, SD, limits of agreement (LoA) with 95% Cis, ICC of K1 K2, and CCT between the Revo Nx, Casia 2 and Galiei G6.**

| | Mean | SD | Range | Difference of the means | SD.for the diff. | Lower endpoint of 95% CI | Upper endpoint of 95% CI | ICC |
|---|---|---|---|---|---|---|---|---|
| K1 Anterior Galilei | 43.155 | 1.390 | 39.43–46.1 | | | | | |
| K1 Anterior Meridian 3mm REVO | 43.210 | 1.557 | 39.5–46.7 | -0.055 | 0.457 | -0.149 | 0.038 | 0.952 |
| K1 Anterior CASIA | 43.191 | 1.374 | 39.5–46.55 | | | | | |
| K1 Anterior Meridian 3mm REVO | 43.210 | 1.557 | 39.5–46.7 | -0.019 | 0.440 | -0.109 | 0.071 | 0.956 |
| K2 Anterior CASIA | 44.177 | 1.385 | 40.62–47.17 | | | | | |
| K2 Anterior Meridian 3mm REVO | 43.987 | 1.531 | 40.2–47.2 | 0.190 | 0.552 | 0.077 | 0.303 | 0.921 |
| K2 Anterior Galilei | 44.150 | 1.376 | 40.3–47.46 | | | | | |
| K2 Anterior Meridian 3mm REVO | 43.987 | 1.531 | 40.2–47.2 | 0.163 | 0.535 | 0.054 | 0.273 | 0.927 |
| K1 Posterior CASIA | -5.986 | 0.217 | -0.93 | | | | | |
| K1 Posterior Meridian 3mm REVO | -5.771 | 0.248 | -1.1 | -0.215 | 0.084 | -0.233 | -0.198 | 0.595 |
| K2 Posterior CASIA | -6.296 | 0.247 | -1.16 | | | | | |
| K2 Posterior Meridian 3mm REVO | -6.027 | 0.271 | -1.5 | -0.269 | 0.087 | -0.287 | -0.251 | 0.532 |
| K1 Posterior Galilei | -6.096 | 0.280 | -1.55 | | | | | |
| K1 Posterior Meridian 3mm REVO | -5.771 | 0.248 | -1.1 | -0.325 | 0.236 | -0.374 | -0.277 | 0.166 |
| K2 Posterior Galilei | -6.530 | 0.392 | -2.12 | | | | | |
| K2 Posterior Meridian 3mm REVO | -6.025 | 0.272 | -1.5 | -0.505 | 0.308 | -0.568 | -0.441 | 0.016 |
| CCT Central Power Galilei | 569.489 | 37.024 | 482–637 | | | | | |
| CCT Central Power CASIA | 545.559 | 35.697 | 474–612 | 23.930 | 9.579 | 21.957 | 25.903 | 0.774 |
| CCT Central Power CASIA | 545.000 | 36.155 | 474–612 | | | | | |
| CCT Central Power REVO | 537.389 | 35.928 | 467–605 | 7.611 | 3.717 | 6.833 | 8.390 | 0.973 |
| CCT Central Power Galilei | 569.128 | 37.577 | 482–637 | | | | | |
| CCT Central Power REVO | 537.389 | 35.928 | 467–605 | 31.739 | 10.322 | 29.577 | 33.901 | 0.653 |

disc tomography with high sped spectral OCT. We believe that the lack of agreement showed in our paper. compared with previous studies showing high agreement. is related to the better precision of modern devices. Orbscan II. for instance. showed ICC. of 0.984 and 0.981 for the flat and step axis respectively while Galilei (a newer device) had an ICC. of 0.991 and 0.994 [12]. Revo NX showed 0.991 and 0.989. It is important to note that our study group was more than 3 times bigger.

Measuring anterior corneal surface is easier than measuring the posterior [21]. In order to measure the latter. sophisticated mathematic algorithms have to be implemented. which is why there is a significant difference between the recordings of the devices. Secondly do to the a very strong reflex at the air/cornea interface makes it difficult to corelcty identify edges. Thirdly posterior surface evaluation is hindered by the errors of the dront surface. Moreover. the size of the posterior measurement is different for all three devices. Casia 2 measures on a 3.2 mm radius while Galilei G6 on a 4 mm and in Revo NX it is within 3 mm. Refractive indexes for posterior or surface can vary in different devices. Anterior surface keratometry can be measured in simulated keratometry when values are calculated from the annular (semi-chord) or in true keratometry where values are measured within the circle. There is no posterior simulated keratometry [22].

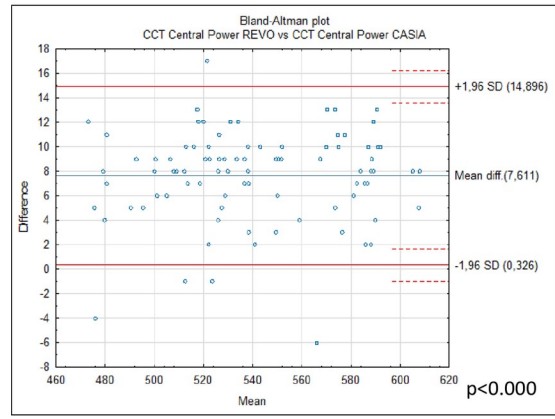

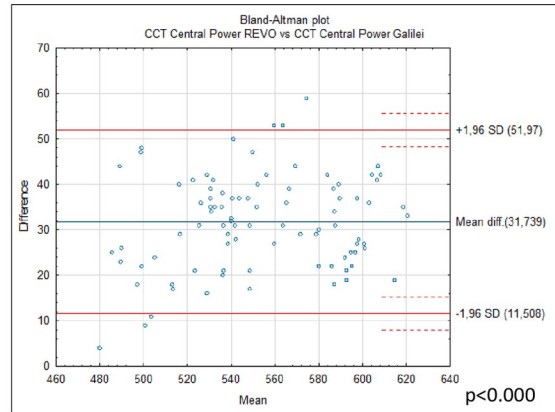

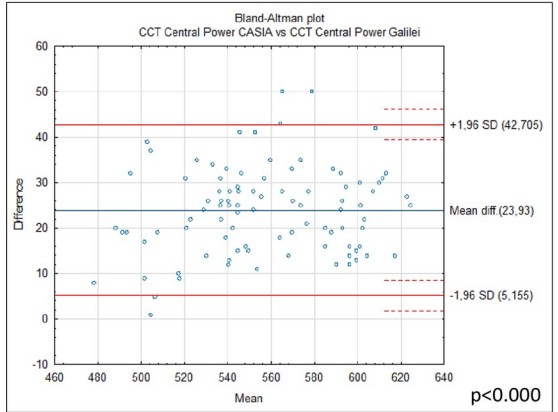

**Fig 4. Bland-Altman plots showing the agreement between central corneal thickness obtained by the Galileli G6.** Casia 2 and Revo NX in 94 normal eyes. The mean difference is represented by the solid blue line whereas the dotted lines represent ±1.96 SD.

## Limitations of this study

Our sample did not include eyes with corneal conditions such as keratoconus or post-transplant where different results might be observed. Secondly. the volunteers were relatively young.

In conclusion. Revo NX provides reliable and repeatable results. Also. inter-operator reproducibility of the measurements is high. The agreement between devices is low and is due to different methods utilized. It is important then not to compare results between devices.

## Supporting information

**S1 File.**
(DOCX)

## Acknowledgments

The authors gratefully acknowledge Prof. Achim Langenbucher from Institute of Experimental Ophthalmology. Saarland University. Homburg/Saar. Germany for his insightful comments about the design of our experiment. We also thank Optopol Technology Ltd. for providing the Revo NX equipment with corneal topography module used in this study. Optopol Technology Ltd. Played no further role in this study.

## Author Contributions

**Data curation:** Adam Wylęgała, Robert Mazur, Bartłomiej Bolek.

**Formal analysis:** Adam Wylęgała, Robert Mazur, Bartłomiej Bolek.

**Investigation:** Bartłomiej Bolek, Edward Wylęgała.

**Methodology:** Robert Mazur.

**Supervision:** Edward Wylęgała.

**Writing – original draft:** Adam Wylęgała.

**Writing – review & editing:** Adam Wylęgała, Robert Mazur, Edward Wylęgała.

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
