## [Decision Letter · Decision Letter 0]

8 Jan 2020

PONE-D-19-32488

Reproducibility, repeatability and agreement of corneal topography measured by Revo NX, Galilei G6 and Casia 2.

PLOS ONE

Dear MD Wylęgała,

Thank you for submitting your manuscript to PLOS ONE. After careful consideration, we feel that it has merit but does not fully meet PLOS ONE’s publication criteria as it currently stands. Therefore, we invite you to submit a revised version of the manuscript that addresses the points raised during the review process.

We would appreciate receiving your revised manuscript by Feb 22 2020 11:59PM. To enhance the reproducibility of your results, we recommend that if applicable you deposit your laboratory protocols in protocols.io, where a protocol can be assigned its own identifier (DOI) such that it can be cited independently in the future. For instructions see: http://journals.plos.org/plosone/s/submission-guidelines#loc-laboratory-protocols

We look forward to receiving your revised manuscript.

Kind regards,

Andrzej Grzybowski

Academic Editor

PLOS ONE

Additional Editor Comments:

Thank you for submitting this interesting. Please revise the ms according to the reviewer's comments.

Journal Requirements:

"Adam Wylęgała is The Kosciuszko Foundation Scholar."

3. Please provide details of the obtained participant consent in the ethics statement on the online submission form. Currently this information is only available in the methods section of your manuscript.

4. Please carefully proofread your manuscript for typographical errors. For example, in the Methods section “… had been informed of the experimental procedure.We included 94 eyes of 35 Males and 59 Females …” should be written as “… had been informed of the experimental procedure. We included 94 eyes of 35 males and 59 females …”

5. Please provide further details regarding how participants were recruited, including the participant recruitment date.

6. We note that you have a patent relating to material pertinent to this article.

a. Please provide an amended statement of Competing Interests to declare this patent (with details including name and number), along with any other relevant declarations relating to employment, consultancy, patents, products in development or modified products etc. Please confirm that this does not alter your adherence to all PLOS ONE policies on sharing data and materials, as detailed online in our guide for authors http://journals.plos.org/plosone/s/competing-interests by including the following statement: "This does not alter our adherence to  PLOS ONE policies on sharing data and materials.” If there are restrictions on sharing of data and/or materials, please state these.

Please note that we cannot proceed with consideration of your article until this information has been declared.

We note that the Revo NX equipment used in this study was provided by Optopol Technology Ltd. Please state this information in your competing interests statement and clarify whether Optopol Technology Ltd. played any further role in the study.

Reviewers' comments:

Reviewer's Responses to Questions

**Comments to the Author**

1. Is the manuscript technically sound, and do the data support the conclusions?

Reviewer #1: Yes

Reviewer #2: Partly

2. Has the statistical analysis been performed appropriately and rigorously? 

Reviewer #1: Yes

Reviewer #2: I Don't Know

3. Have the authors made all data underlying the findings in their manuscript fully available?

Reviewer #1: Yes

Reviewer #2: Yes

4. Is the manuscript presented in an intelligible fashion and written in standard English?

Reviewer #1: Yes

Reviewer #2: Yes

5. Review Comments to the Author

Reviewer #1: In general, this study is very interesting and timely! I strictly recommend publication of this manuscript after a thorough overwork! Here are some comments and recommendations for improvement of the manuscript:

Title:

Title should be reformulated: What do the authors mean with agreement? There should be a statement that measurements were performed in 'normal' eyes

Abstract:

What does 'correct measurements' mean?

Use coreal power of the anterior/posterior surface instead of anterior/posterior keratometry! Keratometry always means anterior surface radius converted to diopters using an artificial keratometer index! Pleas consider this throughout the manuscript!

How were K values averaged? separately for flat and steep meridian?

What are measures such as 0.975 for repeatability stand for? Cronbach's alpha? Is this a value for the steep or the flat meridian?

What is the repeatability for thickness measurement?

Anterior corneal power should be around 49 diopters instead of 43 diopters, maybe the authors converted the front surface radius with a keratometry index instead of reald air-cornea interface data?

Introduction:

Please mention the benefits AND DRAWBACKS of combined posterior/anterio segment OCTs and dedicated anterior segment OCTs! The major drawback of combined systems is the lack of collimated light at the cornea, which means that the measurement results depend on the measurement distance. Another major drawback of all comboned systems is the small diameter of the measurement volume, here 8 mm for the Revo instead of 16 mm for the Casia

Methods:

males and females instead of Males and Females

Did the authors check all volunteers for ectatic diseases such as KC?

6 measurements FROM each operator...

Why didn't the authors record corneal thickness at thinnes point?

Instead of simply evaluating the data of the steep and flat meridian, data should be assessed using vector decomposition, e.g. the classical Humphrey notation. This is necessary due to the fact that beside flat and steep meridian the orientation axes between measurement and devices may vary!

5 µm axial resolution refers to air or medium?

Please clarify: Keratometry...calculated in the 3 mm zone: Where exactly? At a ring with diameter of 3 mm or including all data from the central 3 mm zone? This is important because classical keratometry does not consider the entire 3 mm zone but distinct measurement points at a diameter of around 3 mm depending on the device.

Conversion of radius of curvature to dioptric power using a keratometer index os not appropriate for anterior surface power. Instead, evaluate keratometric power OR anterior and posterior surface power. Keratometric power somehow mimics the behaviour of he entire cornea.

Why is Galilei using posterior mean K instead of evaluating both meridians separately?

What does 'Thus, unlike the (anterior) SimK, it includes the central 1 mm in diameter' mean?

standard deviation instead of Standard deviation...

Results:

Interoperator repeatability and reproducibility??? Sure you refer to reproducibility only comparing the results of different operators...

For all results I recommend to consider vector components instead of flat/steep meridional data due to the fact that otherwise different orientations of 'astigmatism is ignored. Why is corneal thickness missed in tables 1 and 2?

Discussion:

is launched to the market instead of brought to the market

tomographis systems instead of keratometric systems...

Scheimpflug camera instead of Sheimpflug camera.

Please state that it is simply a question of using a specific eye model or assumingany average speed of sound or refractive index if ultrasound, Scheimpflug or OCT (with different wavelength) yield different results for central corneal thickness...

Scheimplfug should read Scheimpflug.

'Measuring anterior corneal surface is easier th 258 an measuring the posterior[21]. In order to measure the latter, sophisticated mathematic algorithms have to be implemented, which is why there is a significant difference between the recordings of the devices.' might be half of the truth!! At the front surface there is a very strong reflex due to air-cornea interface which makes exact and reliable detection of the edge difficult. On the other hand, back surface evaluation (not measurement!!) requires inverse raytracing, which means that all errors of front surface measurement affect back surface measurements!

Figures:

The resolution of the figures is weak. The authors should use other image formats with a higher resolution for the upload of the revised version!

In general, in times where topographers and tomographers are more and more used for diagnostics and surgery planning such evaluations of systems which are newly on the market are more than welcome and very important for the reader! My congratulations to this interesting manuscript!

Reviewer #2: The goal of this study is to compare the repeatability and inter-operator reproducibility of a new corneal topographer module of Revo NX SD-OCT with Galilei G6 Schimpflug camera and Casia 2 SS-OCT in normal eyes.

I suggest several things that could make the presentation clearer.

1. Put the description of the devices before the description of the measurement techniques. Just switch the order.

2. There are several places in the text and in the footnotes of the tables in which punctuation (a comma) is needed. Also, the references in the footnotes do not totally match what they are referring to. For example in Table 1, Standard deviation Sw = within-subject standard.

3. This section below is not clear. Clarify what the 6 measurements are. Also put commas and an “and” in to clarify the paragraph.

Every participant had 6 measurements (for each operator) starting with 3 Topo scan program on the Revo NX carried out by each operator to measure repeatability and reproducibility, followed by one corneal map measurement on Galilei G6, and? Corneal Map scan on Casia 2. For every device, anterior and posterior K1 and K2 values were recorded as well as apical CCT. Only measurements well centered and with high quality indexes were included in the study.

I can only count 5 measurements if you are referring to devices: 3 topo scan, one Galilei GT and one Casia 2.

I can only count 5 measurements if you are referring to variables: anterior and posterior K1 and K2 and CCT.

4. This section needs clarification. You don’t need to capitalize mean and standard deviation. Use a comma to separate items in a list. Note that “Within-subject standard deviation” is just hanging at the end of the paragraph. It is not a sentence.

Numerical results for repeatability and reproducibility contain six quantities computed for observers separately and respectively for the entire dataset: mean, standard deviation, within-subject standard deviation, test-retest repeatability, within-subject coefficient of variation, and intraclass correlation coefficient were calculated for repeatability and reproducibility of the Revo NX. Within-subject standard deviation.

5. The statistical methods needs more elaboration. Define how each of these measurements were computed:

• within-subject standard deviation (Is this the root mean square error from the model?)

• test-retest repeatability

• Cov% (this appears to be with-subject standard deviation divided by the mean rather than the standard deviation divided by the mean) Is this computed using the rmse from the model?)

• ICC—was this computed based on a random effects model? How many measurements were included? If this was done for each operator, was it the 3 measures each?

6. Interoperator and Reproducibility Section and Table 1 issuses:

• It is not clear which devices are being compared in the table? Is is all three? Or just the new device? Based on the comments below the table, it appears to be an assessment of Revo NX. Labels for the table and section would help the reader.

• It is labeled as “interoperator repeatability,” yet the table contains information for each operator (A and B). Therefore, this should be labeled as “intra-operator.”

• Column in table says observator. Observator is not an English word. Use “operator” to match the title of the table.

• I don’t understand how there is an ICC for each operator. If the two operators are being compared, there should be only one ICC.

• OR Are you saying that the 3 measurements for each operator are being compared separately? If that is the case, the title should be ‘Intra-operator.’

• How many measurements were used? This needs to be specified in the table.

• From the text below the table, it appears that the way the operators are being compared is by taking the difference between values in the table. To carry out a true INTER-operator assessment, the measurement from both operators need to be in the model. For example, the data would need to be laid out like this.

Operator Measurement K1 K2…

A 1

A 2

A 3

B 1

B 2

B 3

• Note that the computations from the model need to use the number of repetitions used. In the models that appear in Table 1, I assume that there were 3 measurements per operator. So, in the computation of ICC the number of measurements must be used.

o BET=(MSB-MSE)/3 ; *** NOTE: this denominator must match number of observers/measurements ;

7. Table 2 issues

• I am not sure what is being compared in this table. At least the table title specified that it is for Revo NX. Define the reproducibility. Is this the result of having both operators in the same model?

• The footnote also needs to be cleaned up.

8. Table 3

• What is the rationale for comparing the Galilei G6 to the other two devices?

• If Galilei G6 is the “gold standard,” it would be helpful to state that in the methods.

9. Comparison section of results

• This section is comparing pair-wise differences between devices for paired t-tests. What is a little confusing is whether just one measurement per device is being used. It appears that 3 measures per operator were used for the Revo NX and one measurement for each of the other devices (based on my understanding of the methods). So, which of the three Revo NX measures are being compared with the other devices and for which operator. Or was this done separately for each operator and only results for one operator presented.

• This is just one example of the lack of explanation and clarity shows up in this paper.

10. Plots

• Plots are hard to read. They seem blurry. A better rendering should be done for the manuscript.

• Note that all the plots say “P=0.000.” This should be corrected to match the text that describes the paired t-test or use P<0.001 for labels on plots if the values are actually that low.

• Bland-Altman plots are a visual indication of agreement. In addition to the paired t-tests, the limits of agreement should be mentioned. These are presented in the plots. One would state within which limits one would consider devices to be interchangeable, regardless of the statistical significance of the tests.

11. Overall, this paper may offer new information that is useful. However, clearing up some of the questions above would help readers understand it better.

6. PLOS authors have the option to publish the peer review history of their article (what does this mean?). If published, this will include your full peer review and any attached files.

Reviewer #1: No

Reviewer #2: Yes: Sandra Stinnett

---

## [Author Response · Author response to Decision Letter 0]

25 Feb 2020

Reviewer #1:

Comment: In general, this study is very interesting and timely! I strictly recommend publication of this manuscript after a thorough overwork! Here are some comments and recommendations for improvement of the manuscript:

Title:

Comment:

Title should be reformulated: What do the authors mean with agreement? There should be a statement that measurements were performed in 'normal' eyes

Response: we changed the tile in accordance with the reviewer’s wish.

Reproducibility, and repeatability of corneal topography measured by Revo NX, Galilei G6 and Casia 2 in normal eyes. 

Abstract:

Comment:

What does 'correct measurements' mean?

Response: that the quality index was high enough. We decided to omit this word., 

Comment 

Use coreal power of the anterior/posterior surface instead of anterior/posterior keratometry! Keratometry always means anterior surface radius converted to diopters using an artificial keratometer index! Pleas consider this throughout the manuscript!

Response:

We changed the keratometry to corneal power both within the abstract as within the text. 

Comment:

How were K values averaged? separately for flat and steep meridian?

What are measures such as 0.975 for repeatability stand for? Cronbach's alpha? Is this a value for the steep or the flat meridian?

Reponses:

 The measure value such as 0.975 stand for Intraclass correlation coefficient and 0.975 is the lowest for posterior K2 while 0.994 is for the anterior K1

 The Revo NX showed intraclass correlation coefficient ranging from 0.975 for the posterior K2 surface, and 0.994 for anterior K1. 

Comment:

What is the repeatability for thickness measurement?

Response:

We did calculate it

Central corneal thickness A 531.05 32.54 1.47 4.07 0.28 0.998

 B 530.73 32.68 1.50 4.16 0.28 0.998

Comment:

Anterior corneal power should be around 49 diopters instead of 43 diopters, maybe the authors converted the front surface radius with a keratometry index instead of reald air-cornea interface data?

Response:

Thank you for this comment it is true that that anterior corneal power is around 49 however as we used a commercially available software in our study which automatically divides anterior corneal power through keratometry index. The reason is why anterior surface result gives 43 D instead of 49 D is used refraction index. Refraction index which is used in the software in compared devices is 1.3375 instead of 1.376 (ref index of cornea). the reason is that devices have to comparable to standard keratometers which are using ref index 1.3375. This ref index is demanded by regulation EN ISO 19980:2012 Ophthalmic instruments — Corneal topographers. In point 3.11 and 3.12. The reference device provides all corneal power maps but the aim of study was to compare common parameters. 

Introduction:

Comment:

Please mention the benefits AND DRAWBACKS of combined posterior/anterio segment OCTs and dedicated anterior segment OCTs! The major drawback of combined systems is the lack of collimated light at the cornea, which means that the measurement results depend on the measurement distance. Another major drawback of all comboned systems is the small diameter of the measurement volume, here 8 mm for the Revo instead of 16 mm for the Casia 

Response:

Thank you for this comment. We decided to include this statement in our paper:

The major benefit of using a combined system is the lower price, and higher resolution. While drawback is the lack of collimated light at the cornea, whcihc leads to the measurements being distance dependent. Further the field of view is twice smaller then in the single use device. 

Methods:

Comment

males and females instead of Males and Females

Response: 

We changed the Males and Females to males and females. 

Comment:

Did the authors check all volunteers for ectatic diseases such as KC?

Response: All volunteers were checked for ecstatic disease during scanning. 

Comment:

6 measurements FROM each operator...

Response:

Yes this was corrected

Comment:

Why didn't the authors record corneal thickness at thinnes point?

Instead of simply evaluating the data of the steep and flat meridian, data should be assessed using vector decomposition, e.g. the classical Humphrey notation. This is necessary due to the fact that beside flat and steep meridian the orientation axes between measurement and devices may vary!

Response: This is indeed a very insightful comment, we decided to do that as we based our study on other papers. For our future papers we are going your method.

Comment

5 µm axial resolution refers to air or medium?

Response:

It is 5µm in tissue so medium. 

Comment:

Please clarify: Keratometry...calculated in the 3 mm zone: Where exactly? At a ring with diameter of 3 mm or including all data from the central 3 mm zone? This is important because classical keratometry does not consider the entire 3 mm zone but distinct measurement points at a diameter of around 3 mm depending on the device.

Response:

At a cross of two diameters of a ring both diameters have 3 mm. 

Comment:

Conversion of radius of curvature to dioptric power using a keratometer index os not appropriate for anterior surface power. Instead, evaluate keratometric power OR anterior and posterior surface power. Keratometric power somehow mimics the behaviour of he entire cornea.

Response:

According norm and Calculations of Gullstrand model eye and common use in corneal topographer is presenting anterior surface in power in dioptres instead of millimetres. 

Comment:

Why is Galilei using posterior mean K instead of evaluating both meridians separately?

Reposne:

We tried to contact Zimmer company but received no reply so we cannot deliberate on why it is so. 

Comment:

What does 'Thus, unlike the (anterior) SimK, it includes the central 1 mm in diameter' mean?

standard deviation instead of Standard deviation...

Repose:

We decided to delete this sentence 

Comment:

Results:

Interoperator repeatability and reproducibility??? Sure you refer to reproducibility only comparing the results of different operators...

Response:

Interoperator was changed to intraoperator repeatability as it refers to the differences between one operator, while reproducibility was interoperator meaning we compared dresults from operator A and B. 

Comment

For all results I recommend to consider vector components instead of flat/steep meridional data due to the fact that otherwise different orientations of 'astigmatism is ignored. Why is corneal thickness missed in tables 1 and 2?

Discussion:

Response:

We have once more recalculated the date and included the CCT in tab 1 and 2. 

Comment:

is launched to the market instead of brought to the market

tomographis systems instead of keratometric systems...

Scheimpflug camera instead of Sheimpflug camera.

Response:

We have checked it, and the phrase “bring to market” is commonly used. We corrected the last two. 

https://www.theguardian.com/business/2020/jan/13/polluting-vehicles-could-be-pulled-from-uk-sale-say-carmakers

“Carmakers are rushing to bring to market new electric cars with zero exhaust emissions – including Volkswagen’s ID.3, Vauxhall’s Corsa-e and an electric Fiat 500 – this year, but production will initially be limited as factories gear up. At the same time, they are keen to hang on to their profitable but polluting sales of internal combustion engines.”

Comment:

Please state that it is simply a question of using a specific eye model or assumingany average speed of sound or refractive index if ultrasound, Scheimpflug or OCT (with different wavelength) yield different results for central corneal thickness...

Response:

We have included following statement: Different methods yield different results due to the variuose reference models used such as average speed of sound or refractive index. 

Commnet:

Scheimplfug should read Scheimpflug.

Response:We corrected it

Comment:

'Measuring anterior corneal surface is easier th 258 an measuring the posterior[21]. In order to measure the latter, sophisticated mathematic algorithms have to be implemented, which is why there is a significant difference between the recordings of the devices.' might be half of the truth!! At the front surface there is a very strong reflex due to air-cornea interface which makes exact and reliable detection of the edge difficult. On the other hand, back surface evaluation (not measurement!!) requires inverse raytracing, which means that all errors of front surface measurement affect back surface measurements!

Response we included this statement:

Secondly do to the a very strong reflex at the air/cornea interface makes it difficult to corelcty identify edges. Thirdly posterior surface evaluation is hindered by the errors of the dront surface.

Figures:

The resolution of the figures is weak. The authors should use other image formats with a higher resolution for the upload of the revised version!

In general, in times where topographers and tomographers are more and more used for diagnostics and surgery planning such evaluations of systems which are newly on the market are more than welcome and very important for the reader! My congratulations to this interesting manuscript!

 The goal of this study is to compare the repeatability and inter-operator reproducibility of a new corneal topographer module of Revo NX SD-OCT with Galilei G6 Schimpflug camera and Casia 2 SS-OCT in normal eyes.

I suggest several things that could make the presentation clearer.

Comment 1. Put the description of the devices before the description of the measurement techniques. Just switch the order.

Response 1: Thank you for this comment we have changed that. 

Comment 2. There are several places in the text and in the footnotes of the tables in which punctuation (a comma) is needed. Also, the references in the footnotes do not totally match what they are referring to. For example in Table 1, Standard deviation Sw = within-subject standard.

Response 2 We have changed the punctuation both in the footnotes and in the text. We also changed the abbreviations and unified them across the text and in the footnotes. 

3. This section below is not clear. Clarify what the 6 measurements are. Also put commas and an “and” in to clarify the paragraph.

Every participant had 6 measurements (for each operator) starting with 3 Topo scan program on the Revo NX carried out by each operator to measure repeatability and reproducibility, followed by one corneal map measurement on Galilei G6, and? Corneal Map scan on Casia 2. For every device, anterior and posterior K1 and K2 values were recorded as well as apical CCT. Only measurements well centered and with high quality indexes were included in the study.

I can only count 5 measurements if you are referring to devices: 3 topo scan, one Galilei GT and one Casia 2.

I can only count 5 measurements if you are referring to variables: anterior and posterior K1 and K2 and CCT.

Response 3 : Every participant had 8 measurement 6 on Revo NX 3 for operator A and 3 for operator B, 1 on Galieli and 1 on Casia. . 

Comment 4: This section needs clarification. You don’t need to capitalize mean and standard deviation. Use a comma to separate items in a list. Note that “Within-subject standard deviation” is just hanging at the end of the paragraph. It is not a sentence.

Numerical results for repeatability and reproducibility contain six quantities computed for observers separately and respectively for the entire dataset: mean, standard deviation, within-subject standard deviation, test-retest repeatability, within-subject coefficient of variation, and intraclass correlation coefficient were calculated for repeatability and reproducibility of the Revo NX. Within-subject standard deviation.

Response 4: 

Thank you for this comment we have corrected this sentence 

“Numerical results for repeatability and reproducibility contain six quantities computed for observers separately and respectively for the entire dataset: mean, standard deviation (SD.), within-subject standard deviation (Sw.), test-retest repeatability (TRT.), within-subject coefficient of variation (CoV.), intraclass correlation coefficient (ICC.) were calculated for repeatability and reproducibility of the Revo NX.” 

5. The statistical methods needs more elaboration. Define how each of these measurements were computed:

• within-subject standard deviation (Is this the root mean square error from the model?)

• test-retest repeatability

• Cov% (this appears to be with-subject standard deviation divided by the mean rather than the standard deviation divided by the mean) Is this computed using the rmse from the model?)

• ICC—was this computed based on a random effects model? How many measurements were included? If this was done for each operator, was it the 3 measures each?

Response5: 

We have decided to add description of the data as a supporting material. 

Numerical results for repeatability resp. reproducibility contain six quantities computed for observers separately resp. for the entire dataset:

• Mean

• Standard deviation

• Sw

• TRT

• CoV[%]

• ICC

Mean is the arithmetic mean of input values.

Standard deviation is the sample standard deviation, ie. with N-1 in the denominator, where N is the sample size.

Sw = within-subject standard deviation, is the root mean square of sample standard deviations of values measured on a single object, ie. 

Sw = ((σ12+...+σM2)/M)1/2,

where M is the number of objects (eyes) and σk equals the sample standard deviation of values measured on the k-th object.

TRT = test-retest repeatability, is defined as = 2,77·Sw.

CoV = within-subject coefficient of variation, is defined as = Sw/Mean or = 100·Sw/Mean when reported as %.

ICC = intraclass correlation coefficient, is defined as the ratio of appropriate estimated variances. For this, it is assumed that there is a set of measured values yij for the i-th object in j-th repetition; i = 1, 2, ..., N (N = the number of objects) and j = 1, 2, ..., Ni (different numbers of repetitions are permitted for different objects). The measured values are modelled by the equation below: 

yij = μ + σAei + σBeij,

where μ is the average value and e's are independent realizations of a standard normal random variable and σA2 and σB2 are resp. interclass and intraclass variances. ICC is given by:

ICC = sA2/(sA2 + sB2),

where sA2, sB2 are estimated values of the variances σA2, σB2 according to the equations:

ΣiΣj(yij - yi)2 = (M - N)sB2,

Σi(yi - y)2 = (N - 1)(sA2 + sB2/H),

where yi denotes the mean value for i-th object: yi = Σjyij/Ni, while y denotes the overall mean: y = Σiyi/N. In the above, M = ΣiNi equals the total number of measurements and H = N/Σi(1/Ni) is the harmonic mean of Ni's.

Comment 6. Interoperator and Reproducibility Section and Table 1 issuses:

• It is not clear which devices are being compared in the table? Is is all three? Or just the new device? Based on the comments below the table, it appears to be an assessment of Revo NX. Labels for the table and section would help the reader.

• It is labeled as “interoperator repeatability,” yet the table contains information for each operator (A and B). Therefore, this should be labeled as “intra-operator.”

• Column in table says observator. Observator is not an English word. Use “operator” to match the title of the table.

• I don’t understand how there is an ICC for each operator. If the two operators are being compared, there should be only one ICC.

• OR Are you saying that the 3 measurements for each operator are being compared separately? If that is the case, the title should be ‘Intra-operator.’

• How many measurements were used? This needs to be specified in the table.

• From the text below the table, it appears that the way the operators are being compared is by taking the difference between values in the table. To carry out a true INTER-operator assessment, the measurement from both operators need to be in the model. For example, the data would need to be laid out like this.

Operator Measurement K1 K2…

A 1

A 2

A 3

B 1

B 2

B 3

• Note that the computations from the model need to use the number of repetitions used. In the models that appear in Table 1, I assume that there were 3 measurements per operator. So, in the computation of ICC the number of measurements must be used.

o BET=(MSB-MSE)/3 ; *** NOTE: this denominator must match number of observers/measurements ;

Response 6: 

• This table is about repeatability of Revo NX and we changed title accordingly.

• We changed the title from inter- to intraoperator

• We changed Observator to Operator. 

• We have applied two ICCs because the values are calculated from 3 meassurements by each operator. Tab 2 shows ICC of 6 meassurements from two operators. 

• Yes that is correct we updated title accordingly. 

• Yes, this has been updated in the title. 

• We agree however this information is provided in the tab 2. 

• We have included the number of repetitions in the title. 

Comment 7. 

Table 2 issues

• I am not sure what is being compared in this table. At least the table title specified that it is for Revo NX. Define the reproducibility. Is this the result of having both operators in the same model?

• The footnote also needs to be cleaned up.

Response 7. 

• We changed the tile to “Table 2 Revo NX reproducibility based on six measurement from both operators”.

• The foot note has been cleaned up

Comment 8. Table 3

• What is the rationale for comparing the Galilei G6 to the other two devices?

• If Galilei G6 is the “gold standard,” it would be helpful to state that in the methods.

Reposne 8. Galiei is a Scheimflug technology that is longer used and is considered as a gold standard in keratometric measurements. “Scheimpflug technology is considered gold standard in corneal measurements.” 

Comment 9. Comparison section of results

• This section is comparing pair-wise differences between devices for paired t-tests. What is a little confusing is whether just one measurement per device is being used. It appears that 3 measures per operator were used for the Revo NX and one measurement for each of the other devices (based on my understanding of the methods). So, which of the three Revo NX measures are being compared with the other devices and for which operator. Or was this done separately for each operator and only results for one operator presented.

• This is just one example of the lack of explanation and clarity shows up in this paper.

Response 9:

• A mean from 6 measurements from Revo NX was used for comparison between the results. 

• We are sorry for this we hope that the changes implemented are sufficient and make the manuscript easier to read. 

10. Plots

• Plots are hard to read. They seem blurry. A better rendering should be done for the manuscript.

• Note that all the plots say “P=0.000.” This should be corrected to match the text that describes the paired t-test or use P<0.001 for labels on plots if the values are actually that low.

• Bland-Altman plots are a visual indication of agreement. In addition to the paired t-tests, the limits of agreement should be mentioned. These are presented in the plots. One would state within which limits one would consider devices to be interchangeable, regardless of the statistical significance of the tests.

Response 10:

• We changed the figures into higher resolution. 

• We included new table 4 with the LoA values and ICC. 

11. Overall, this paper may offer new information that is useful. However, clearing up some of the questions above would help readers understand it better.

---

## [Decision Letter · Decision Letter 1]

4 Mar 2020

Reproducibility, and repeatability of corneal topography measured by Revo NX, Galilei G6 and Casia 2 in normal eyes.

PONE-D-19-32488R1

Dear Dr. Wylęgała,

We are pleased to inform you that your manuscript has been judged scientifically suitable for publication and will be formally accepted for publication once it complies with all outstanding technical requirements.

With kind regards,

Andrzej Grzybowski

Academic Editor

PLOS ONE

Additional Editor Comments (optional):

Thank you for preparing a high quality revision and including all reviewers comments. Thank you also for submitting your interesting paper to our journal.

Reviewers' comments:

Reviewer's Responses to Questions

**Comments to the Author**

1. If the authors have adequately addressed your comments raised in a previous round of review and you feel that this manuscript is now acceptable for publication, you may indicate that here to bypass the “Comments to the Author” section, enter your conflict of interest statement in the “Confidential to Editor” section, and submit your "Accept" recommendation.

Reviewer #1: All comments have been addressed

2. Is the manuscript technically sound, and do the data support the conclusions?

Reviewer #1: Yes

3. Has the statistical analysis been performed appropriately and rigorously? 

Reviewer #1: Yes

4. Have the authors made all data underlying the findings in their manuscript fully available?

Reviewer #1: Yes

5. Is the manuscript presented in an intelligible fashion and written in standard English?

Reviewer #1: Yes

6. Review Comments to the Author

Reviewer #1: Thank you for this detailled overwork of the manuscript! In the present version this manuscript is ready for publication!

7. PLOS authors have the option to publish the peer review history of their article (what does this mean?). If published, this will include your full peer review and any attached files.

Reviewer #1: No

---

## [Editor Report · Acceptance letter]

9 Mar 2020

PONE-D-19-32488R1 

Reproducibility, and repeatability of corneal topography measured by Revo NX, Galilei G6 and Casia 2 in normal eyes. 

Dear Dr. Wylęgała:

I am pleased to inform you that your manuscript has been deemed suitable for publication in PLOS ONE. Congratulations! Your manuscript is now with our production department. 

With kind regards,

on behalf of

Dr. Andrzej Grzybowski 

Academic Editor

PLOS ONE